# VISION TRANSFORMER WITH IRREGULAR ATTENTION

## ABSTRACT

Compression of Transformer is a natural request that arose in the computer vision community. Apart from quantization that hardly rely on hardware, sparsification is another way to remove redundant parts, usually based on mask training or sparsity regularization. We propose the novel compressed structure of multi-head self-attention (MHSA) mechanism called Irregular Attention (IrrAtt). IrrAtt is built on top of BTD-(L,L,1) tensor decomposition and is aimed at sparsifying pre-trained Vision Transformer by pruning query and key (QK) contraction dimension in MHSA block. We derive the algorithm of rank selection procedure for BTD-(L,L,1) based on the structure of fusion layer obtained from CP decomposition of original MHSA kernels. In order to improve the compression ratio with least possible quality loss we introduce the fine-tuning schemes that yield each head its own sub-optimal rank for QK in the IrrAtt. We validated the proposed scheme for DeiT architectures on ILSVRC-2012 dataset. Our results show that IrrAtt has better performance compared to original MHSA compressed by SVD. It indicates that attention heads have non-uniform importance and require different QK contract dimensions.

## 1 INTRODUCTION

Since their arrival, transformers Vaswani et al. (2017) became main neural architecture for the natural language processing and then they have been successfully applied for different computer vision tasks Dosovitskiy et al. (2021); Touvron et al. (2021); Carion et al. (2020); Strudel et al. (2021). Despite their great success, this architecture is significantly computationally consuming, which makes applying it on mobile devices a challenging task. For example, the DeiT-Base model, which is a typical vision Transformer (ViT) model, requires 17.6 GFLOPs and 86M parameters for a single forward pass. Most of the parameters are concentrated in the linear layers of MHSA and Multilayer perceptron (MLP).

While the MLP compression is well-studied topic and can be performed by conventional methods, the compression of MHSA is relatively new and underexplored. Moreover, MHSA has a more complicated structure, as it consists of several linear layers that interact with each other. Therefore, MHSA compression demands novel methods that take into account its peculiarities.

Several methods have been proposed to improve the efficiency of the self-attention mechanism. One way is to restrict the attention region of each token to local or windowed attention Liu et al. (2021b); Zhang & Yang (2021). However, this approach sacrifices the global self-attention and requires many layers to enlarge the receptive field. Another way is to reduce the number of tokens Tang et al. (2021); Hou & Kung (2022); Rao et al. (2021) or dimensions of tokens Zhu et al. (2021); Yu et al. (2022b); Chen et al. (2021). While token pruning shows good results in image classification, its performance on dense downstream tasks may be limited. Token channel dimension can be reduced with pruning or low-rank approximation.

This paper is focused on the compression of MHSA. We propose a new Vision Transformer by extending the MHSA with regular attention weights to the new Irregular Attention (*IrrAtt*), whose ranks for heads can be automatically determined, while the total ranks of heads is bounded. The contributions of this paper are summarized as follows:

- A new attention *IrrAtt* layer based on BTD-$(L, L, 1)$ decomposition, where each attention weight can have own rank.

- Rank selection procedure for *IrrAtt* block based on rank identification from the structure of the fusion layer in *CP SlimAtt*.
- Initialization procedure for *IrrAtt* block based on BTD-$(L, L, 1)$ ALS algorithm.

To demonstrate the efficiency and applicability of the proposed method, it is tested on DeiT and *ILSVRC-2012* Deng et al. (2009) dataset.

The rest of the paper is organized in the following way. Section 5 provides an overview of neural network compression and decomposition rank selection literature.

## 2 PRELIMINARIES

### 2.1 NOTATION

We use the following conventions to denote tensors and tensor operations. We use bold calligraphic letters to represent tensors, such as $\mathcal{Y} \in \mathbb{R}^{I_1 \times I_2 \times \cdots \times I_N}$, where $N$ is the order of the tensor, and $I_n$ is dimension of the mode-$n$. For example, a order 2 tensor $\mathbf{W} \in \mathbb{R}^{I \times J}$ is a matrix of size $I \times J$. Following Cichocki et al. (2016), we represent tensors graphically by circles, while each outgoing leg from a circle represents the indices of a specific mode (see Figure 1). An order-$N$ tensor $\mathcal{Y} \in \mathbb{R}^{I_1 \times I_2 \times \cdots \times I_N}$, which has order $N$ and size $I_n$, is represented by a circle with $N$ legs, each of size $I_n$, $(n = 1, 2, ..., N)$. An interconnection between two circles denotes a contraction of tensors.

We use superscripts to distinguish different tensors/matrices of the same type or shape, such as $\mathbf{W}^Q, \mathbf{W}^K$, and $\mathbf{W}^V$ for the query, key, and value weight matrices in a transformer layer. We use subscripts to index slices or elements of a tensor, such as $\mathbf{W}_h$ for the $h$-th slice (matrix) of an order-3 tensor $\mathcal{W}$, or $\mathbf{W}(i, j)$ for the $(i, j)$-th element of a matrix $\mathbf{W}$. We use colon ":" to indicate all elements along a mode, such as $\mathbf{W}(i, :)$ for the $i$-th row of a matrix $\mathbf{W}$.

The symbol "∘" denotes tensor outer product. Tensor unfolding (or matricization) of $\mathcal{W}$ reshapes a tensor into a matrix by arranging the elements along the mode-$n$ and is denoted by $\mathbf{W}_{(n)}$. For example, unfolding an order-3 tensor $\mathcal{W}$ along mode 2, written as $\mathcal{W}_{(2)}$, gives a matrix of size $I_2 \times I_1 I_3$.

### 2.2 MULTI-HEAD SELF-ATTENTION

Attention mechanisms are widely used to model the interactions between different inputs. The operation for a single self-attention head is defined as:

$$\mathbf{A}_h = \text{SoftMax}\left(\frac{\mathbf{Q}_h \mathbf{K}_h^\top}{\sqrt{d}}\right) \mathbf{V}_h \in \mathbb{R}^{N \times D}, \tag{1}$$

and the outcome of MHSA is given by

$$\mathbf{O} = \sum_{h=1}^{H} \mathbf{A}_h \mathbf{X} \mathbf{W}_h^V (\mathbf{W}_h^O)^\top, \tag{2}$$

where $\mathbf{Q}_h = \mathbf{X} \mathbf{W}_h^Q, \mathbf{K}_h = \mathbf{X} \mathbf{W}_h^K$, and $\mathbf{V}_h = \mathbf{X} \mathbf{W}_h^V$. $\mathbf{W}_h^Q, \mathbf{W}_h^K, \mathbf{W}_h^V \in \mathbb{R}^{F \times D}$ are (Q)uery, (K)ey, and (V)alue matrices, respectively, which project input sequence $\mathbf{X} \in \mathbb{R}^{N \times F}$; $N$ - sequence length, $F$ - sequence element feature size, $D$ - attention feature size. The tensor diagram of MHSA is illustrated in Figure 1. MHSA complexity is $4NFDH + 2HN^2 D$, which is quadratic to input sequence length $N$ and attention feature length $D$.

### 2.3 TRUNCATT

Multiplication of Q and K- projections in MHSA could be seen as contraction of input $\mathbf{X}$ and joint query-key (QK) weight tensor $\mathcal{Y}^{QK}$ in the following way:

$$\mathbf{A}_h^* = \mathbf{Q}_h \mathbf{K}_h^\top = \mathbf{X} \mathcal{Y}^{QK}(:, :, h) \mathbf{X}^\top,$$

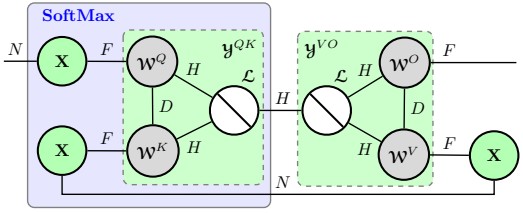

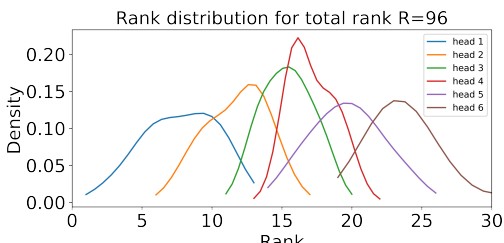

Figure 1: The tensor diagram of MHSA. $\mathbf{X}$ is an input matrix with sequence length $N$ and feature size $F$. Gray nodes $\mathcal{W}^i$ for $i = \{Q, K, O, V\}$ are weights and a white node $\mathcal{L}$ is a super-diagonal tensor. $\mathcal{Y}^{QK}$ and $\mathcal{Y}^{VO}$ are contraction of three tensors inside green shaded boxes. **SoftMax** is applied to the tensor network inside the blue box.

Figure 2: BTD-$(L, L, 1)$ rank distribution for total rank 96. BTD-$(L, L, 1)$ ranks for each block are sorted and split into $H = 6$ groups. Then, density for each group (head) is visualized.

where each head $h$ has its own attention weight, $\mathcal{Y}^{QK}(:, :, h)$ of size $F \times F$, and $\mathcal{A}^*$ is an attention tensor before softmax normalization. Reconstruction of this weight tensor from $\mathcal{W}^Q$ and $\mathcal{W}^K$ weights will increase the computational complexity of QK part in MHSA. On the contrary, one can compress the joint QK weight tensor $\mathcal{Y}^{QK}$ by approximating the head attention matrices with its low-rank counterparts, e.g., using truncated singular value decomposition (SVD).

We name this layer *TruncAtt*, which stands for truncated attention. The original MHSA needs $NDH(2F + N)$ multiplications to get attention tensor $\mathcal{A}^*$, where $D$ is the QK- contract length (rank). *TruncAtt* can achieve significant savings in terms of multiplications when $R$ is much smaller than $D$.

### 2.4 CP SLIMATT

Instead of compressing attention weight matrix, $\mathcal{Y}^{QK}(:, :, h)$ for each head individually, an alterative method is compress the whole attention weight tensor, $\mathcal{Y}^{QK}$, by the canonical polyadic (CP) model, like in compression of convolutional layers . The new layer leverages the joint subspace among the weight matrices, $\mathbf{W}_h^Q$ and $\mathbf{W}_h^K$, and obtains common single projection matrices, $\mathbf{S}^Q$ and $\mathbf{S}^K$ shared among all heads. This way, we replace the original QK weights, $\mathcal{Y}^{QK}$, with a new architecture that has fewer parameters and lower computational complexity, while preserving the performance of the layers.

$$\mathcal{Y}^{QK} \approx \sum_{r=1}^{R} \mathbf{S}^Q(:, r) \circ \mathbf{S}^K(:, r) \circ \mathbf{S}^H(:, r). \tag{3}$$

We name this the *CP SlimAtt* layer. The new structure consists of three matrices: $\mathbf{S}^Q$ and $\mathbf{S}^K$, both of size $F \times R$, and $\mathbf{S}^H$, of size $H \times R$ as a fusion or multi-head expansion. The new layer returns output for each head, $h = 1, 2, \ldots, H$, as

$$\mathbf{A}_h^* = \underbrace{\mathbf{X}\mathbf{S}^Q}_{Q-projection} \underbrace{\mathrm{diag}(\mathbf{S}^H(h, :))}_{head\ expansion} \underbrace{(\mathbf{S}^K)^\top \mathbf{X}^\top}_{K-projection}. \tag{4}$$

The complexity of this step is $\mathcal{O}(NFR)$, where $N$ is the number of tokens, $F$ is the sequence element feature size, and $R$ is the rank of the projection matrices. $\mathbf{Q} = \mathbf{X}\mathbf{S}^Q$, and $\mathbf{K} = \mathbf{X}\mathbf{S}^K$ are Q and K-projections, respectively. The multi-head fusion is performed by multiplying the Q and K-projections with diagonal matrix, $\mathrm{diag}(\mathbf{S}_h^H)$. This requires $N(N + 1)HR$ multiplications, where $H$ is the number of heads.

The rank-$R$ of the CP decomposition controls the trade-off between the accuracy of the approximation and the complexity of the layer. A higher rank-$R$ leads to a better approximation of $\mathcal{Y}^{QK}$, but also increases the complexity.

# 3 MHSA WITH IRREGULAR ATTENTION

## 3.1 SQUEEZING THE CP-SLIMATT

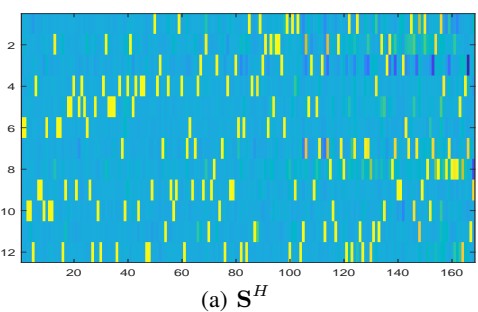

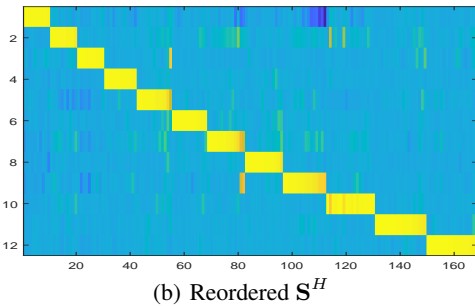

(a) $\mathbf{S}^H$          (b) Reordered $\mathbf{S}^H$

Figure 3: Fusion weight $\mathbf{S}^H$ in the CP-SlimATT layer. (a) finetuned weights, (b) finetuned weights after reorder and rescaling columns in the fusion matrix.

**Remark (From weight sparsity to simplification of CP SlimAtt).** *In this remark, we show how the sparsity structure of the fusion weights in CP SlimAtt can be exploited to simplify the model and reduce its complexity.*

*In Figure 3, we illustrate the third factor matrix, $\mathbf{S}^H$, in DeiT-B with CP SlimAtt layers with rank of 168 fine-tuned on ILSVRC-12. The matrix, $\mathbf{S}^H$, in Figure 3(a) exhibits sparsity structure of the fusion weights. Since CPD is non-unique up to permutation and scaling [1], we can change the order and scale of columns in factor matrices, so that the largest absolute values in each column are 1. After reordering columns and rows (heads) of $\mathbf{S}^H$, Figure 3(b) reveals a very special block-diagonal structure of the fusion layer: 12 blocks with different sizes, each corresponding to one head. Sparsifying the off-diagonal coefficients in the fusion weights can squeeze the CPD layer to a more compact layer.*

We assume the fusion weights $\mathbf{S}^H$ of size $H \times R$, after row (heads) and columns reordering, can be well approximated by $H$ blocks,

$$\mathbf{S}^H = \mathtt{blkdiag}(\boldsymbol{l}_1, \ldots, \boldsymbol{l}_H) = [\boldsymbol{e}_1 \boldsymbol{l}_1, \ldots, \boldsymbol{e}_H \boldsymbol{l}_H], \tag{5}$$

where each $\boldsymbol{l}_h$ is a row vector of length $D_h$, $D_1 + D_2 + \cdots + R_H = R$. We partition the weight matrices, $\mathbf{S}^Q$ and $\mathbf{S}^K$, into $H$ sub-matrices as

$$\mathbf{S}^Q = \left[\mathbf{S}_1^Q, \ldots, \mathbf{S}_H^Q\right],$$

$$\mathbf{S}^K = \left[\mathbf{S}_1^K, \ldots, \mathbf{S}_H^H\right],$$

where $\mathbf{S}_h^Q$ and $\mathbf{S}_h^K$ are of size $F \times D_h$. The CP-SlimATT layer in (3) becomes a new model comprising $H$ terms of rank-$D_h$

$$\boldsymbol{\mathcal{Y}}^{QK} = \sum_{h=1}^{H} \sum_{r=1}^{D_h} \mathbf{S}_h^Q(:,r) \circ \mathbf{S}_h^K(:,r) \circ \boldsymbol{e}_h \boldsymbol{l}_h(r)$$

$$= \sum_{h=1}^{H} (\mathbf{S}_h^Q \, \mathrm{diag}(\boldsymbol{l}_h) \, (\mathbf{S}_h^K)^\top) \circ \boldsymbol{e}_h$$

$$= \sum_{h=1}^{H} \left( \tilde{\mathbf{S}}_h^Q \left( \tilde{\mathbf{S}}_h^K \right)^\top \right) \circ \boldsymbol{e}_h \tag{6}$$

where $\tilde{\mathbf{S}}_h^Q = \mathbf{S}_h^Q \mathrm{diag}(\boldsymbol{l}_h)^{\frac{1}{2}}$, $\tilde{\mathbf{S}}_h^K = \mathbf{S}_h^K \mathrm{diag}(\boldsymbol{l}_h)^{\frac{1}{2}}$. Concatenate the vectors $\boldsymbol{e}_h$ into a matrix of size $H \times H$ gives a new fusion matrix $\mathbf{S}^H$ which is of much smaller size than the fusion matrix in CP-SlimATT. Here we can relax the structure of $\mathbf{S}^H$ as an identity matrix. $\tilde{\mathbf{S}}_h^Q \left( \tilde{\mathbf{S}}_h^K \right)^\top$ is an equivalent low-rank representation of $\mathbf{S}_h^Q \, \mathrm{diag}(\boldsymbol{l}_h) \, (\mathbf{S}_h^K)^\top$.

---

**Algorithm 1** Initialize BTD-$(L, L, 1)$ Decomposition.

---

**Input:** tensor $\mathcal{X}$ of size $(I_1 \times I_2 \times I_3)$, total rank $R$.
**Output:** BTD-$(L, L, 1)$ factors $[\mathbf{A}, \mathbf{B}, \mathbf{C}]$ with sizes $(I_1 \times R), (I_2 \times R), (I_3 \times I_3)$, respectively; BTD-$(L, L, 1)$ ranks.
**begin**
$\quad$ $\mathbf{A}, \mathbf{B}, \mathbf{C} \leftarrow$ CPD-EPC$(\mathcal{X}, R)$
$\quad$ Normalize $\mathbf{C}$ so that largest absolute values in each column are 1
$\quad$ $\boldsymbol{c}_{max}$, order $\leftarrow \max(\mathbf{C}_1, \text{axis} = 1)$ # find max values and their indices over CP rank dim.
$\quad$ $\mathbf{A}, \mathbf{B}, \boldsymbol{c}_{max} \leftarrow$ permute$(\mathbf{A}, \text{order}), (\mathbf{B}, \text{order}), (\boldsymbol{c}_{max}, \text{order})$
$\quad$ Split $\mathbf{C}$ into $I_3$ blocks $\boldsymbol{c}_{R_i}^i = [c_1^i \ldots c_{R_i}^i]$ for $i \in \overline{1, I_3}$ to get ranks $(R_1, \ldots, R_{I_3})$.
$\quad$ $\mathbf{C} \leftarrow \mathbf{I}$, where $\mathbf{I}$ is identity matrix
**end**

---

We call the squeezed model of CP-SlimAtt in (6) *TruncAtt* with irregular contract dimensions $D_h$. The Irregular Attention (*IrrAtt*) block has complexity of *TruncAtt* layer and performance of *CP SlimAtt* layer.

To conclude, we have shown how to simplify the CP SlimAtt model by exploiting the sparsity structure of the fusion weights. We next show methods to obtain the IrrAtt layers in which ranks of attention weights can be automatically defined. Good initialization of weights in IrrAtt layers is essential to prevent big accuracy drop of the new network before finetuning the model.

### 3.2 FROM CP-SLIMATT TO BTD IRR-ATT

The first method to initialize *IrrAtt*'s weights is from the weights in *CP SlimAtt* by seeking block-diagonal of the fusion weight matrices, $\mathbf{S}^H$. Ranks, $D_h$, of the attention weight matrices in *IrrAtt* are the lengths of blockterms, $\boldsymbol{l}_h$. We summarize the proposed procedure in Algorithm 3 for weight initialization, multiple block term decomposition in Algorithm 3 which exploits the single block term decomposition in Algorithm 2.

To obtain the weights in the *CP-SlimAtt*, we used the sensitivity aware algorithm that ensures a stable forward pass, namely CPD-EPC Phan et al. (2020) for *CP SlimAtt*. We note that ordinary algorithms like ALS could also be used for tensor decomposition, but they may not guarantee stability. To get weights for *IrrAtt*, the procedure described in Algorithm 3 was used. Firstly, BTD-$(L, L, 1)$ components are initialized using Algorithm 1. Then, each BTD-$(L, L, 1)$ component is updated at a time using Algorithm 2.

### 3.3 IRR-ATT WITH RANK-CONSTRAINTS

An alternative approach which can reveal the ranks, $D_h$, in *IrrAtt* is to constraint the fusion weight matrices, $\mathbf{S}^H$ in *CP SlimAtt* to have non-overlapping weights due to block-diagonal structure, and non-negativity constraints since $\mathbf{S}^H$ comprises singular values of the weight attention matrices. The proposed loss can be written as

$$\min_{\theta} \quad \mathcal{L}_{train}(\theta), \quad \text{s.t.} \quad \mathbf{S}_i^H \geq 0, \quad \mathbf{S}_i^H (\mathbf{S}_i^H)^\top = \mathbf{I}, i = 1, \ldots, N_{\text{blocks}} \quad (7)$$

### 3.4 RANK SELECTION

Tensor and matrix decompositions, like SVD and CPD, have hyperparameter, *rank*, which describes the computational complexity of the decomposition. BTD-$(L, L, 1)$ has its own rank for each component. The local compression approach was used, where each attention block had the same target compression ratio.

$$\frac{\text{FLOPs}(R)}{\text{FLOPs}_{orig}} \leq \text{CR},$$

where $R$ denotes either rank (SVD or CPD) or total rank (BTD-$(L, L, 1)$ ), FLOPs$_{orig}$ is FLOPs of the original model, CR is the target compression ratio. In this case, the rank for SVD, CPD,

---

**Algorithm 2** Fit Single BTD-$(L, L, 1)$ Component.

---

**Input:** tensor $\mathcal{X}$ of size $(I_1 \times I_2 \times I_3)$, the rank $R$, and maximum number of iterations *maxiter*.
**Output:** BTD-$(L, L, 1)$ component factors $[\mathbf{A}, \mathbf{B}, \boldsymbol{c}]$.
**begin**
    Initialize $\boldsymbol{c}_i$ as leading eigenvector of $\mathcal{X}_{(3)}\mathcal{X}_{(3)}^\top$
    **for** $j = 1, \ldots,$ *maxiter* **do**:
        Rank-$R$ truncated SVD of $\mathcal{X} \times_3 \boldsymbol{c}^\top \approx \mathbf{A}\mathbf{B}^\top$
        $\mathbf{T} \leftarrow \mathcal{X}_{(3)}$ khatri-rao$(\mathbf{A}, \mathbf{B}^\top)$
        Best rank-1 approximation to $\mathbf{T} \approx s\,\boldsymbol{c}\boldsymbol{d}^\top$
    **end**
    $\mathbf{A} \leftarrow \mathbf{A}\,\mathrm{diag}(s\boldsymbol{d})$
**end**

---

**Algorithm 3** BTD-$(L, L, 1)$ Decomposition with CPD-EPC Initialization

---

**Input:** tensor $\mathcal{X}$ of size $(I \times J \times H)$, total rank $R$.
**Output:** BTD-$(L, L, 1)$ factors $[\mathbf{A}, \mathbf{B}, \mathbf{C}]$ with sizes $(I \times R), (J \times R), (H \times H)$, respectively; BTD-$(L, L, 1)$ ranks.
**begin**
    Get factors $([\mathbf{A}_1, \ldots, \mathbf{A}_{I_3}], [\mathbf{B}_1, \ldots, \mathbf{B}_{I_3}], [\boldsymbol{c}_1, \ldots, \boldsymbol{c}_{I_3}])$ and ranks $(R_1, \ldots, R_{I_3})$ using Algorithm 1.
    **for** $j = 1, \ldots,$ *maxiter* **do**:
        **for** $i = 1, \ldots, I_3$ **do**:
            $\mathcal{X}' \leftarrow \mathcal{X} - \sum_{k \neq i}(\mathbf{A}_k, \mathbf{B}_k, \boldsymbol{c}_k)$
            $\mathbf{A}_i, \mathbf{B}_i, \boldsymbol{c}_i \leftarrow$ fit\_component$(\mathcal{X}', R_i)$ using Algorithm 2
        **end**
    **end**
**end**

---

or BTD-$(L, L, 1)$ was determined based on the target compression ratio and was the same for all attention blocks, so different compressed attention blocks could be compared.

## 4 EXPERIMENTS

### 4.1 EXPERIMENTAL SETUP

The experiments were conducted with the efficient neural network framework Pytorch on a GPU server with one NVIDIA® Tesla® A100 GPU, AMD EPYC 7452 32-Core CPU, and 80 GB RAM. A pre-trained DeiT-Base and DeiT-Small model shipped with Torchvision and Timm library was used as a baseline for the *ILSVRC-12*. DeiT-Base included 86.57M parameters and 17.6B FLOPs, while DeiT-Small included 22M parameters and 4.6B FLOPs. In our approach, the original model was compressed with selected tensor decomposition and then fine-tuned for 60 epochs with batch size 256 (DeiT-B) or 512 (DeiT-S) using an optimizer parameters from original DeiT pipeline. Top-1 accuracy was estimated on the *ILSVRC-12* validation set. One fine-tune pass (60 epochs) took 1.5-2 GPU days on a single NVIDIA® Tesla® A100. Summary of optimizer parameters is presented in Appendix A.3.

### 4.2 MODEL COMPRESSION RESULTS

Our goal was to compare *TruncAtt*, *CP SlimAtt* and *IrrAtt* in terms of FLOPs, number of parameters and top-1 accuracy drop. We first applied weight decomposition to the QK block, and then fine-tuned the network. VO blocks were compressed with *TruncAtt*, while MLP was compressed using SVD. All results, which are discussed below, are presented in Table 1.

### 4.3 INITIALIZATION FOR IRRATT

We have compared results for *IrrAtt* with initialization using BTL-$(LL1)$, constrained CP. In addition, the constrained CP gives initialization that is further from original local minima compared to BTD-$(L, L, 1)$ ; hence, this results in lower top-1 accuracy for constrained CP initialization.

### 4.4 IRRATT VS CP SLIM

The experiments with DeiT models have shown that *IrrAtt* has top-1 accuracy close to *CP SlimAtt*, while the FLOPs CR is significantly lower, $\geq 1.3$ times. The results supports the observation that fusion layer in *CP SlimAtt* has low-rank structure.

### 4.5 IRRATT VS TRUNCATT

*TruncAtt* is a specific case of *IrrAtt*, when all contract dimensions have the same size. *IrrAtt* has shown slightly higher FLOPs CR (due to extra head fusion layer), while achieving better top-1 accuracy, plus 0.15 - 0.2%. This result proves the head slices in joint QK weight $\mathcal{Y}^{QK}$ have different rank. Moreover, we took *IrrAtt* with total rank 96 for each block, sorted the component ranks for each BTD-$(L, L, 1)$ and split them into $H = 6$ groups, i.e., smallest rank came to the first group while largest to the last group. The rank density distribution is presented in Figure 2. Hence, in all attention blocks, the joint QK weight $\mathcal{Y}^{QK}$ has both: low-rank and high-rank slices.

Table 1: Comparison of different *TruncAtt*, *CP SlimAtt* and *IrrAtt* blocks for DeiT on the *ILSVRC-12* validation set. The Top-1 Acc column is the model's top-1 accuracy. CR stands for compression ratio. *FLOPs CR* is estimated without accounting SoftMax and GELU modules. *MHSA FLOPs CR* contains FLOPs only for MHSA block. VO blocks are compressed with *TruncAtt*, while MLP is compressed with SVD. Blank MLP rank stands for a model without MLP compression. [†] denotes *IrrAtt* initialization with constrained CP.

| QK block | QK rank | VO rank | MLP rank | Top-1 Acc, % | MHSA FLOPs CR, % | FLOPs CR, % | # param CR, % |
|---|---|---|---|---|---|---|---|
| | | | | DeiT-Small ($H = 6$) | | | |
| Original | 64 | 64 | | 79.85 | 100 | 100 | 100 |
| *TruncAtt* | 16 | 32 | | 78.87 (-0.98) | 37.5 | 76.2 | 79.9 |
| *TruncAtt* | 22 | 32 | | 79.00 (-0.85) | 42.2 | 78.0 | 81.4 |
| *TruncAtt* | 32 | 32 | | 79.19 (-0.61) | 50.0 | 81.0 | 83.9 |
| *CP* | 96 | 32 | | 79.16 (-0.69) | 50.3 | 81.1 | 79.9 |
| *CP* | 132 | 32 | | 79.10 (-0.75) | 59.7 | 84.7 | 81.4 |
| *CP* | 192 | 32 | | 79.33 (-0.52) | 75.5 | 90.7 | 84.0 |
| *IrrAtt* | 96 | 32 | | 79.03 (-0.82) | 38.5 | 76.6 | 79.9 |
| *IrrAtt* | 132 | 32 | | 79.18 (-0.67) | 43.1 | 78.4 | 81.4 |
| *IrrAtt* | 192 | 32 | | 79.45 (-0.40) | 51.0 | 81.4 | 83.9 |
| [†]*IrrAtt* | 192 | 32 | | 78.96 (-0.89) | 51.0 | 81.4 | 83.9 |
| Original | 64 | 64 | 152 | 78.45 (-1.40) | 100 | 69.4 | 67.6 |
| *TruncAtt* | 32 | 32 | 152 | 77.35 (-2.60) | 50.0 | 50.4 | 51.5 |
| *CP* | 192 | 32 | 152 | 77.49 (-2.36) | 75.5 | 60.1 | 51.5 |
| *IrrAtt* | 192 | 32 | 152 | 77.58 (-2.26) | 51.0 | 50.8 | 51.5 |
| | | | | DeiT-Base ($H = 12$) | | | |
| Original | 64 | 64 | | 81.80 | 100 | 100 | 100 |
| *TruncAtt* | 13 | 32 | 304 | 80.53 (-1.27) | 35.2 | 44.7 | 45.7 |
| *CP SlimAtt* | 168 | 32 | 304 | 80.84 (-0.96) | 49.7 | 49.9 | 46.0 |
| *IrrAtt* | 168 | 32 | 304 | 80.74 (-1.06) | 37.0 | 45.4 | 46.0 |

Table 2: Comparison of *IrrAtt* and channel pruning methods for attention block of DeiT-Base on the *ILSVRC-12* validation set. The $\Delta$ Top-1 Acc column is the model's top-1 accuracy drop with regard to the original model. CR is compression ratio.

| Model | $\Delta$ Top-1 Acc, % | FLOPs CR, % | # param CR, % |
|---|---|---|---|
| VTP Zhu et al. (2021) | -1.1 | 56.8 | 55.5 |
| WDPruning Yu et al. (2022a) | -1.06 | 56.6 | 65.0 |
| UVC Yu et al. (2022b) | -1.23 | 45.5 | - |
| *QK IrrAtt(168) + VO IrrAtt(32)* (our) | -1.06 | **45.4** | **46.0** |

## 4.6 COMPARISON WITH CHANNEL PRUNING

*IrrAtt* was compared with the channel pruning method Zhu et al. (2021); Yu et al. (2022b;a), which removes input and/or output channels in a linear layer. Our approach had a smaller accuracy drop at lower FLOPs CR level, as shown in Table 2.

## 4.7 MLP COMPRESSION

Compression of MLP is not the major aim of this paper. It can be done with the truncated SVD, i.e., replacing each linear layer with a sequence of two linear layers.

## 5 RELATED WORKS

Neural network compression is a long-established field of science where several main techniques have been suggested to strike a balance between the efficiency and speed of pre-trained models. Among these techniques are pruning, quantization, knowledge distillation, low-rank approximation and others.

### 5.1 NEURAL NETWORK COMPRESSION AND ACCELERATION

Our proposed framework can be mostly related to low-rank methods but also have some connections with pruning. Thus we only make a brief overview of these two family of methods.

#### 5.1.1 PRUNING

Neural network weight pruning has established that not all weights carry equal significance. Hence, commonly more than half can be eliminated without affecting performance. For Transformers, three pruning techniques are widely used: unstructured pruning or weight sparsification (removing individual weights), structured pruning (eliminating a group of weights) and token/patch pruning, i.e. discarding sequence elements.

In Zhu et al. (2021) ViT was trained with sparsity regularization, which involved incorporating dimension-wise sparsity through the pruning of dimensions in linear projections. Yu et al. Yu et al. (2022b) in their work suggested an optimization framework that took into account budget constraints. Their framework enables the joint learning of model weights and layer-wise pruning ratios with masks and skip configurations. Rao et al. Rao et al. (2021) employed a lightweight prediction module to dynamically select a subset of the most relevant tokens. This module is integrated into multiple layers of ViT, allowing hierarchical sparsification. Consequently, the number of pruned tokens progressively increases with each prediction module. In contrast, Patch Slimming Tang et al. (2021) implements a top-down approach to remove redundant patches. Initially, effective patches in the last layer are identified, and they are subsequently utilized to guide the patch selection process in previous layers. Hou et al. Hou & Kung (2022) introduced a pruning criterion based on statistical dependency. This criterion utilizes the Hilbert-Schmidt norm of the cross-variance operator and can be applied to various dimensions. Consequently, it is capable of identifying and removing detrimental components across heads, features, and sequence dimensions simultaneously.

### 5.1.2 Low-rank Approximation

To reduce the complexity of neural network models without altering the network structure, a common approach is to employ low-rank approximation of weights Novikov et al. (2015); Ou et al. (2023). Singular Value Decomposition (SVD) Chen et al. (2021) and its more advanced version Fisher-Weighted SVD (FWSVD) Hsu et al. (2022), are naturaly used for ViTs, which mostly consist of linear layers represented as matrices.

When it comes to more advanced scenarios, matrix tensorization followed by tensor decomposition is a popular approach. Tensor Train (TT) decomposition Oseledets (2011), including Matrix Product Operator (MPO), are widely chosen in this scenario. TT decomposition has been successfully applied to tensorized linear layers in various domains such as language processing Liu et al. (2021a); Li et al. (2022) and speech recognition He & bin Zhong (2019). TT is also utilized for ViT in image classification Minh et al. (2022) and object detection Zhen et al. (2022) tasks. In Liu et al. (2021a) Liu et al. show that MPO outperforms Canonical Poliadic decomposition (CPD) Harshman (1970); Hillar & Lim (2013) and Tucker decomposition (TKD) Tucker (1963). TKD has been employed to compress all layers of the BERT model simultaneously Ren et al. (2022). An alternative is Tensor Chain (or Tensor Ring) decomposition (TC), which approximates the weights of linear and convolutional layers represented as high-order tensors Wang et al. (2018).

An advanced approach in the Tensorized Transformer Ma et al. (2019) involves modeling the interactions between query, key and value tensors in MHSA using Block Term Decomposition (BTD) De Lathauwer (2008). Furthermore, Tuformer Liu et al. (2022) introduces method for joint learnable weights across heads through weight matrix reparameterization.

### 5.2 Decomposition Rank Selection

The choice of rank for low-rank methods balances the compression ratio and quality drop values. In most cases the task of finding the optimal rank is a complex and computationally demanding task. Thus establishing the procedure of optimal rank selection is important part of low-rank methods. In Sobolev et al. (2022) authors propose a Proxy-based Automatic tensor Rank Selection method (PARS) that utilizes a Bayesian optimization approach to find the best combination of ranks for neural network compression.

## 6 Conclusion

This work proposes *IrrAtt*, which has the same computation complexity as *TruncAtt* and performance similar to *CP SlimAtt*. Utilizing the sparse structure of a fusion layer in *CP SlimAtt*, a powerfull technique to mitigate rank determination for heads in *IrrAtt* and to initialize the weights inside query-key (QK) block with BTD-$(L, L, 1)$ factors is presented. *IrrAtt* highlights the non-uniform structure inside QK block in MHSA and showed good compression properties. Yet this non-uniform structure requires some extra low-level optimization for the efficient implementation of MHSA to utilize modern hardware efficiently. For the next steps we are planning to develop further the rank minimization procedures in order to achieve better compression ratios at the same quality drop.

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

## A  APPENDIX

### A.1  ABBREVIATIONS

The following abbreviations are used in this manuscript:

| | |
|---|---|
| ViT | Vision Transformer |
| CNN | Convolutional Neural Network |
| MHSA | Multi-Head Self-Attention |
| MLP | Multilayer Perceptron |
| ILSVRC-12 | ImageNet |
| SVD | Singular Value Decomposition |
| CPD | Canonical Polyadic Decomposition |
| CPD-EPC | Canonical Polyadic Decomposition with Error-Preserving Correction |
| GELU | Gaussian Error Linear Units |
| QK | Query-Key |
| VO | Value-Output |
| CR | Compression Ratio |
| FLOPs | Number of Floating Point Operations |
| MACs | Number of Multiply–Add Operations |

### A.2  TENSOR DIAGRAM REPRESENTATION

Following Cichocki et al. (2016), we represent tensors graphically by nodes (circles), while each outgoing line (leg) from a node represents the indices of a specific mode (see Figure 1). In our adopted notation, each vector, matrix or tensor is represented by a circle, while the order of a tensor is determined by the number of lines (legs) connected to it. An order-$N$ tensor $\mathcal{Y} \in \mathbb{R}^{I_1 \times I_2 \times \cdots \times I_N}$, which has order $N$ and size $I_n$, is represented by a circle with $N$ legs, each of size $I_n$, $(n = 1, 2, ..., N)$. An interconnection between two nodes denotes a contraction of tensors. More illustrations of tensor networks can be found in Supplementary.

#### TENSOR DECOMPOSITIONS

In the following section we briefly describe tensor decompositions used in the paper.

#### A.2.1  CANONICAL POLIADIC DECOMPOSITION (CPD)

The Canonical Poliadic Decomposition (CPD) Harshman (1970); Hillar & Lim (2013) represents an order-N tensor by the sum of rank-1 tensorsor equivalently by factor matrices interconnected through a super-diagonal tensor. For an order-3 tensor $\mathcal{Y}$ of size $I_1 \times I_2 \times I_3$ with CP rank $R$, the CPD has the form:

$$\mathcal{Y} \simeq \sum_{r=1}^{R} \boldsymbol{a}_r \circ \boldsymbol{b}_r \circ \boldsymbol{c}_r,$$

where $\mathbf{A} = [\boldsymbol{a}_1, \ldots, \boldsymbol{a}_R]$, $\mathbf{B} = [\boldsymbol{b}_1, \ldots, \boldsymbol{b}_R]$, and $\mathbf{C} = [\boldsymbol{c}_1, \ldots, \boldsymbol{c}_R]$ are factor matrices of size $I_1 \times R, I_2 \times R,$ and $I_3 \times R$, respectively. The illustration for CPD of an order-3 tensor is presented in Figure 4.

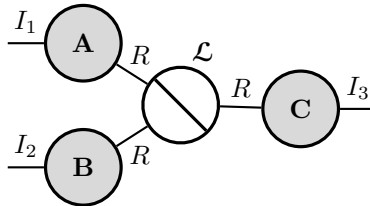

Figure 4: CP decomposition of an order-3 tensor.

## A.2.2 TUCKER DECOMPOSITION

Comparing to the CP decomposition, the Tucker decomposition (TKD) provides a more flexible factorization of an order-$N$ tensor into a relatively small size core tensor and factor matrices, given in the form as

$$\mathcal{Y} \simeq \sum_{r_3=1}^{R_3} \sum_{r_2=1}^{R_2} \sum_{r_1=1}^{R_1} g_{r_1 r_2 r_3} \left( \boldsymbol{a}_{r_1} \circ \boldsymbol{b}_{r_2} \circ \boldsymbol{c}_{r_3} \right) = \mathcal{G} \times_1 \mathbf{A} \times_2 \mathbf{B} \times_3 \mathbf{C},$$

where $\mathcal{Y} \in \mathbb{R}^{I_1 \times I_2 \times I_3}$ is the given data tensor, $\mathcal{G} \in \mathbb{R}^{R_1 \times R_2 \times R_3}$ is the core tensor, and $\mathbf{A} = [\boldsymbol{a}_1, \dots, \boldsymbol{a}_{R_1}]$, $\mathbf{B} = [\boldsymbol{b}_1, \dots, \boldsymbol{b}_{R_2}]$, and $\mathbf{C} = [\boldsymbol{c}_1, \dots, \boldsymbol{c}_{R_3}]$ are the factor (component) matrices. The core tensor (typically, $R_n \ll I_n$) models a potentially complex pattern of mutual interaction between the vectors in different modes. The illustration for TKD of an order-3 tensor is presented in Figure 5.

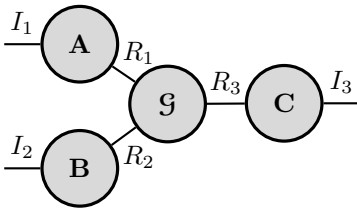

Figure 5: Tucker decomposition of an order-3 tensor.

## A.2.3 BLOCK-TERM DECOMPOSITION (BTD)

Block-term decomposition (BTD) De Lathauwer (2008) is a hybrid of CPD and TKD, and it models the data as the sum of multiple Tucker terms,

$$\mathcal{Y} \simeq \sum_{t=1}^{T} \mathcal{G}_t \times_1 \mathbf{A}_t \times_2 \mathbf{B}_t \times_3 \mathbf{C}_t,$$

where $\mathcal{G}_t$ are order-3 core tensors, while $\mathbf{A}_t$, $\mathbf{B}_t$ and $\mathbf{C}_t$ are factor matrices. BTD comprises a smaller number of parameters than TKD. So far, there are no available proper selection criteria of the block size (rank of BTD) and the number of terms.

## A.3 TRAINING DETAILS

The compressed model was fine-tuned for 60 epochs with batch size 256 using an optimizer with parameters presented in Table 3.

## A.4 TRUNCATT AS CP SLIMATT

We show *TruncAtt* as a special case of *CP SlimAtt* with constraints on the fusion matrix, $\mathbf{S}^H$. Assume the attention matrix $\mathbf{Y}_h$ in *TruncAtt* is represented as rank-$D$ truncated SVD

$$\mathbf{Y}_h = \mathbf{U}_h \operatorname{diag}(\boldsymbol{\lambda}_h) \mathbf{V}^T . \tag{8}$$

Table 3: Optimizer parameters for fine-tuning DeiT-Base model.

| Parameter | Value |
|---|---|
| optimizer | AdamW |
| lr | 5e-05 |
| eps | 1e-08 |
| momentum | 0.9 |
| weight decay | 0.0 |
| scheduler | Cosine |
| min lr | 1e-06 |
| warm-up lr | 1e-06 |
| warm-up epochs | 5 |
| model EMA | True |

We concatenate matrices $\mathbf{U}_h$ into a matrix $\mathbf{S}^Q$, matrices $\mathbf{V}_h$ into $\mathbf{S}^K$, and build the fusion matrix $\mathbf{S}^H$ as block diagonal of singular values, $\boldsymbol{\lambda}_h$, i.e.,

$$\mathbf{S}^Q = [\mathbf{U}_1, \ldots, \mathbf{U}_H],$$
$$\mathbf{S}^K = [\mathbf{V}_1, \ldots, \mathbf{V}_H],$$
$$\mathbf{S}^H = \texttt{blkdiag}(\boldsymbol{\lambda}_1, \ldots, \boldsymbol{\lambda}_H),$$

By this way, $\mathbf{S}^Q$, $\mathbf{S}^K$ and $\mathbf{S}^H$ form an equivalent *CP SlimAtt* layer.

