# OpenReview forum: "Vision Transformer with Irregular Attention"
_ICLR.cc/2024/Conference — ICLR 2024 Conference Withdrawn Submission_

### Official Review · Reviewer_ycW6 · 2023-10-31

**Soundness:** 3 good
**Presentation:** 2 fair
**Contribution:** 3 good
**Rating:** 5
**Confidence:** 3

**Summary:**

This paper mainly focuses on the compression of Multi-Head Self-Attention (MHSA) mechanism. This paper introduces a novel compressed structure for MHSA named Irregular Attention (IrrAtt). This structure is built upon the BTD-(L,L,1) and aims to sparsify pre-trained Vision Transformers by pruning the query and key (QK) contraction dimension in the MHSA block. This paper also presents an algorithm for rank selection based on the structure of the fusion layer derived from the CP decomposition of original MHSA kernels. The main goal is to achieve better compression ratios without compromising the quality of the model.

**Strengths:**

1. Irregular Attention (IrrAtt) provides a new perspective on compressing the Multi-Head Self-Attention mechanism, especially for computer vision tasks. It is applicable to pretrained Transformer models, and holds substantial research value in the field of Machine Learning system and model deployment.
2. The Vision Transformer can be significantly sparsified by using BTD-(L,L,1) tensor decomposition for constructing IrrAtt, which results in a more compact model. It theoretically holds certain feasibility, and is thoroughly discussed.
3. The proposed rank selection algorithm, derived from the fusion layer structure, enables each attention head to have its optimal rank for the Query-Key (QK) contraction.
4. The experiments have validated that the proposed method can achieve a good balance between performance maintenance and model compression.

**Weaknesses:**

1. The paper presents a number of conceptual and technical difficulties, and the ambiguous explanations make it challenging for readers to understand the paper. The introduction of the methods is not detailed enough, making it hard to grasp the true contributions of the author.
2. It may be necessary to carefully adjust the rank for the QK contraction in the IrrAtt for each attention head in order to achieve optimal performance.
3. The proposed rank selection algorithm, derived from the fusion layer structure, may result in extra computing overhead, particularly when the model is being trained.
4. Although validation on a single dataset can provide valuable insights for research, it's beneficial for the model's robustness and generalization to be validated on multiple datasets.
5. The paper did not conduct ablation experiments on various modules of the method, such as initialization, making it difficult for me to judge its true effectiveness.
6. There are several mistakes in the article, such as the MHSA formula in Equation 2. Please clarify or provide references if my understanding is incorrect. English abbreviations appearing for the first time in the paper should be followed by their full names or explanations.

**Questions:**

1. Are the comparison methods TruncAtt and CP SlimAtt first introduced in this paper? If not, please indicate the reference. If they are, there is a lack of comparison with existing methods, making it difficult to evaluate the level achieved by the proposed method.
2. How is the compression ratio controlled to reach the target compression ratio in this method? Are the significant discrepancies in the FLOPs and params CR values in Table 1 due to different target compression ratios set by the comparison methods?
3. The current version leaves me doubting its reproducibility.

---

> ### Author Response · Authors · 2023-11-16
>
> We want to thank the Reviewer. We will revise the manuscript significantly to incorporate the comments and suggestions of the Reviewers. We have addressed the comments about comparison with other existing compression methods, and about ablation study in the comment to all reviewers.
>
> Finding the optimal rank for tensor decomposition is a complicated task. This paper used a local compression approach in which each transformer block has the same compression ratio. Still, the proposed method simplifies the rank search procedure as with the given total rank of MHSA (the rank of CP SlimAtt), we get a rank for each head based on the block diagonal structure of a fusion layer.
>
> The proposed rank selection algorithm needs either extra decomposition or fine-tuning with regularization. Still, the decomposition could be done in parallel and needs no forward and backward passes, while fine-tuning with regularization may take only a few iterations.
>
> The proposed method is for post-training compression. In this scenario, the smaller model is initialized with weights obtained after the approximation of weights in the original model. Hence, there is no option to change the initialization for the model.
>
> We thank the Reviewer and will correct Equation (2), which has an extra $XW^V_h$ in the sum. It should be $O = \sum_{h=1}^H W^O_h A_h$. To get this, we transform equation $O = W^O \times \text{concat} (A_1, \dots, A_H)$
> into $O = \sum_{h=0}^{H-1} W^O(:, Dh+1:D(h+1))A_h = \sum_{h=1}^{H} W^O_h A_h$.
>
> The compression ratio is controlled by the choice of the rank for a decomposition. IrrAtt, TruncAtt, and CP SlimAtt each have different elements; thus, the dependency between FLOPs and the number of parameters varies among compression methods.
>
> The results are reproducible. We use DeiT checkpoints from the official PyTorch repository and fine-tune a pipeline similar to DeiT one with some changes described in Experimental Setup (Section 4.1). Besides, CPD with different initializations gives a similar structure, close to the block diagonal one, like the one shown in Figure 3.

---

### Official Review · Reviewer_v2e6 · 2023-11-01

**Soundness:** 2 fair
**Presentation:** 2 fair
**Contribution:** 2 fair
**Rating:** 3
**Confidence:** 3

**Summary:**

This paper proposes a mechanism called Irregular Attention to compress the retrained Vision Transformer, which is built based on BTD-(L,L,1) tensor decomposition. The proposed method automatically determines the own rank of attention weight under the constraint of the total ranks of heads.

**Strengths:**

The paper proposes a method which has the same computation complexity as TruncAtt and performance similar to CP SlimAtt.

The paper provides quantitative results to prove the effectiveness of the proposed method.

**Weaknesses:**

The paper is not very well written. For example, the abbreviation "CP" has made multiple prior appearances without prior explanation, only being clarified in Section 4.2, which may confuse the reader.

The paper evaluates the effectiveness of the proposed method based on the experimental results on DeiT and ILSVRC-2012 dataset, which is not comprehensive. The paper should conduct experiments on more datasets to have a more solid conclusion.

**Questions:**

Please refer to the weakness part.

---

> ### Author Response · Authors · 2023-11-16
>
> We thank the Reviewer for the comments.
> We have addressed the comments about comparison with other existing compression methods, and about experiments with other models and datasets in the comment to all reviewers.

---

### Official Review · Reviewer_QrK6 · 2023-11-01

**Soundness:** 3 good
**Presentation:** 2 fair
**Contribution:** 2 fair
**Rating:** 3
**Confidence:** 2

**Summary:**

This paper aims to compress Transformers. It proposes the Irregular attention build on top of BTD-(L,L,1). It sparsifies pre-trained vision transformers by pruning the query and key contract dimensions in the MHSA. A fine-tuning scheme is also introduced to improve the performance. The proposed irregular attention is validated for DeiT on ILSVRC-2012 dataset. Experiments show the better results of the proposed method.

**Strengths:**

The studied problem of compressing vision transformers is quite important for the community.

The idea of diversing the importance of multi-attention heads in transformers is reasonable.

Using BTD-(L,L,1) in this problem is new.

**Weaknesses:**

The paper only evalutes on one dataset and one vision transformer. It needs more validations to support the paper's arguments.

The organization of experiments can be improved. Instead of presenting the results, it is better also to analyze the results and give the readers insights about the improvement if possible.

It seems that the paper did not compare to other compression methods. The methods compared in Table 1 are the preliminaries for the proposed one, however, many related works described in Section 5 are not compared.

The contributions listed in the end of Section 1 are not significant enough for an ICLR paper.

The importance of the obtained results and the derived method need to be further strengthed.

**Questions:**

Is there any evidence to support the assumption of (5)?

The way of presenting Algorithm 1-3 should be revised to improve its readability.

It lacks an overview of the proposed irregular attention, and how it can be used in existing vision transformer architectures.

---

> ### Author Response · Authors · 2023-11-16
>
> We want to thank the Reviewer. We will revise the manuscript significantly to incorporate the comments and suggestions of the Reviewers. We have addressed the comments about comparison with other existing compression methods, and about experiments with other models and datasets in the comment to all reviewers.
>
> Regarding the importance of the proposed method, we aimed to study different properties of the proposed method. We took DeiT model as one of the simplest and most widely used vision transformer models in order to show the efficiency and flexibility of our approach. Thus, we decided to concentrate on DeiT to study our method deeply.
>
> The assumption (5) is supported by the structure of weights in CP SlimAtt layers for DeiT, one of which is illustrated in Figure 3.
>
> The proposed method is general and could be straightforwardly applied to many vision transformers, as it replaces multiplication between $W^Q$ and $W^Q$, and between $W^V$ and $W^O$ with their low-rank counterparts and does not modify other parts.

---

### Official Review · Reviewer_NApC · 2023-11-01

**Soundness:** 3 good
**Presentation:** 2 fair
**Contribution:** 3 good
**Rating:** 3
**Confidence:** 4

**Summary:**

This paper proposes Irregular Attention (IrrAtt) for the compression of multi-head self-attention (MHSA) in vision transformers. IrrAtt is built on top of BTD-(L,L,1) tensor decomposition and is aimed at sparsifying pre-trained vision transformers by pruning the query and key contraction dimension in the MHSA block. The proposed IrrAtt is validated for the DeiT architecture on the ILSVRC-2012 dataset.

**Strengths:**

The compression of vision transformers is an important research problem. This paper is well-motivated. The writing is professional and convincing.

**Weaknesses:**

The comparison with existing methods is very limited, i.e., only three methods are compared. Considering that there has been a very large literature on the compression and efficient design of vision transformers, such a limited comparison cannot demonstrate the effectiveness of the proposed method.

The proposed IrrAtt is only validated for the DeiT architecture. Considering that there have been many popular transformer architectures such as Swin Transformer, PVT, and MViT, the only validation for DeiT cannot demonstrate the effectiveness of the proposed method.

There is no ablation study in this paper. This paper has many designs and components (Eq. (1) – Eq. (7), Alg. 1 - Alg. 3), and it is important and necessary to evaluate each of these designs and components. Recently, ablation study is also a necessary part of computer vision papers, especially for deep learning papers.

Will the code be released? This is not mentioned in the paper. This is important to ensure the reproducibility of the paper.

**Questions:**

Please see the above weaknesses.

---

> ### Author Response · Authors · 2023-11-16
>
> We thank the Reviewer for the comments.
> We have addressed the comments about comparison with other existing compression methods, about experiments with other models and datasets, and about ablation study in the comment to all reviewers. We want to consider the comment about the ablation study in more detail. Equations (1-2) describe the original MHSA, while Equations (3-4) express the Canonical Poliadic decomposition and how it can be applied to MHSA. Equations (1-3)  are known from the literature, and Equation (4) is their implication. Equation (5) formulates our assumption about block diagonal structure for the fusion layer, and Equation (6) shows how CPD could be converted to BTD-(L,L,1) using Equation (6). Constraint in Equation (7) is a modification of the original training loss, and it is a simple way to turn the fusion layer into a block diagonal one. We are going to release the codes.

---

### Author Response · Authors · 2023-11-16

We appreciate the reviewer's concern that the comparison with existing compression methods is limited. However, we would like to point out that the domain of neural network compression is very broad and diverse, and it is not feasible to compare our method with every possible approach. Our method is designed to find a sparse representation of MHSA and is closely related to structural pruning. Therefore, we have compared our method with the most recent and relevant structural pruning methods in Table 2. We also note that our method is compatible with other compression techniques, such as quantization and token pruning, and we plan to explore their combination in future work. Other low-rank methods mentioned in Related Works are either applied to different domains, such as NLP, or tested on simpler datasets, such as CIFAR-10/FashionMNIST, and thus are not suitable for comparison. However, to address the reviewer's comment, we will include more pruning methods in the revised version of the paper, which were omitted in the original submission due to space constraints.

We justify our choice of focusing on DeiT architecture and ImageNet dataset for several reasons. First, image classification on ImageNet is a challenging task that requires high performance and robustness from the model, and thus exposes the sensitivity of the model to the compression method. This is not always the case for smaller and easier datasets, such as CIFAR-10 or FashionMNIST. Second, DeiT architecture is a good representative of vision transformers, as it is based on a vanilla Transformer from NLP. As our method only modifies the multiplication between $W^Q$ and $W^K$, and between $W^V$ and $W^O$ in MHSA block, it can be easily applied to many vision transformers without affecting their original structure. Third, DeiT mainly consists of MHSA and MLP, which makes it easier to isolate and evaluate the effect of our method, as the result is not influenced by other non-compressed components, such as in object detection or image segmentation. Finally, most of the existing pruning methods for vision transformers presented in Table 2 and in the literature were also tested on DeiT architecture and ImageNet dataset, which allows for a fair and consistent comparison.

Our paper aims to study different aspects of our method, such as rank selection and initialization of the IrrAtt layer. For rank selection, we use a local compression approach (see Section 3.4) as optimal rank search would require a large number of simulations. For initialization of the IrrAtt layer, we experiment with two cases: initialization with constrained CP or initialization with BTD-(L,L,1) decomposition. The former case assumes fine-tuning with constraint (see Equation (7)), while the latter one is presented in Algorithms (1-3): initialization, single component update, and overall structure. Initially, we used SDF [1] to obtain BTD-(L,L,1) decomposition, but it was much slower, and we had to switch to our proposed method. The comparison of the initializations is discussed in Section 4.3. We believe that this constitutes a sufficient ablation study, but we did not highlight it explicitly in the text. We will revise the paper to make the ablation study more clear and visible.

[1] Sorber, L., Barel, M.V., and Lathauwer, L.D. (2015). Structured Data Fusion. IEEE Journal of Selected Topics in Signal Processing, 9, 586-600.